# ABBEL: Learning Natural-Language Belief States for Memory-Efficient Interaction

## Abstract

As the time horizons of sequential decision-making tasks grow, keeping full interaction histories in model context becomes increasingly costly. Recent work reduces context lengths by instead conditioning decision-making agents on recursively updated natural-language summaries, which are concise and interpretable. However, they underperform agents with access to the full context, suggesting that they fail to generate sufficient summaries. To address this we propose ABBEL, a recursive summarization framework that isolates and directly supervises each summary's information contents in the form of explicit natural-language *belief states*. First, we analyze the belief states generated by frontier models under ABBEL across five domains, and verify that performance is often degraded due to omitting or incorrectly updating information. We also discover settings where models use memory inefficiently by retaining extraneous information. We target these limitations by fine-tuning with two RL-based methods: *belief grading*, which reduces update errors by rewarding belief generations based on their information content, and *peak belief penalties*, which encourage compressing the beliefs with the greatest memory footprints. We demonstrate that these methods significantly reduce the performance gap with full context models, and enable ABBEL to outperform prior memory agent work by 40% while using 67% of the memory. Our code is available at `https://anonymous.4open.science/r/optimal-explorer-dev-CC6A/`.

## 1 Introduction

Complex sequential decision-making tasks such as software development and scientific research require hundreds or thousands of steps of environment interaction, often exceeding the practical limits of even frontier and long context models (Liu et al., 2024). We propose **ABBEL** (Acting through Belief Bottlenecks Expressed in Language), a framework for training LLMs to maintain compact, interpretable contexts through multi-step interaction in the form of explicit, recursively updated natural-language belief states.

Recent work shows LLMs can be effectively trained with end-to-end reinforcement learning fine-tuning (RLFT) to recursively update natural-language context summaries through multi-step interaction (Zhou et al., 2026). However, performance drops below models trained with full context access (Cassano & Rush, 2026), suggesting that the agents do not learn to preserve sufficient information in the summaries for optimal decision making. One possible reason is that the sparse outcome rewards used in end-to-end RL provide only weak supervision for learning what information should be retained in intermediate summaries (Lightman et al., 2024).

To provide more direct supervision and diagnostics, ABBEL isolates the summaries' information contents into explicit *belief states*, in contrast to prior approaches that entangle memory and reasoning within each summary (MEM1, Zhou et al., 2026). ABBEL combines a rollout framework (Fig. 1a) with RL-based methods for targeted belief supervision to improve performance and memory efficiency (Fig. 1c). On each interaction step, ABBEL's rollout framework first prompts the agent to generate a posterior belief conditioned on the prior belief and last step (Fig. 1a, *update belief*), and then prompts for an action conditioned on only the posterior (Fig. 1a, *select action*). The belief prompts are domain-agnostic, leveraging pretrained knowledge

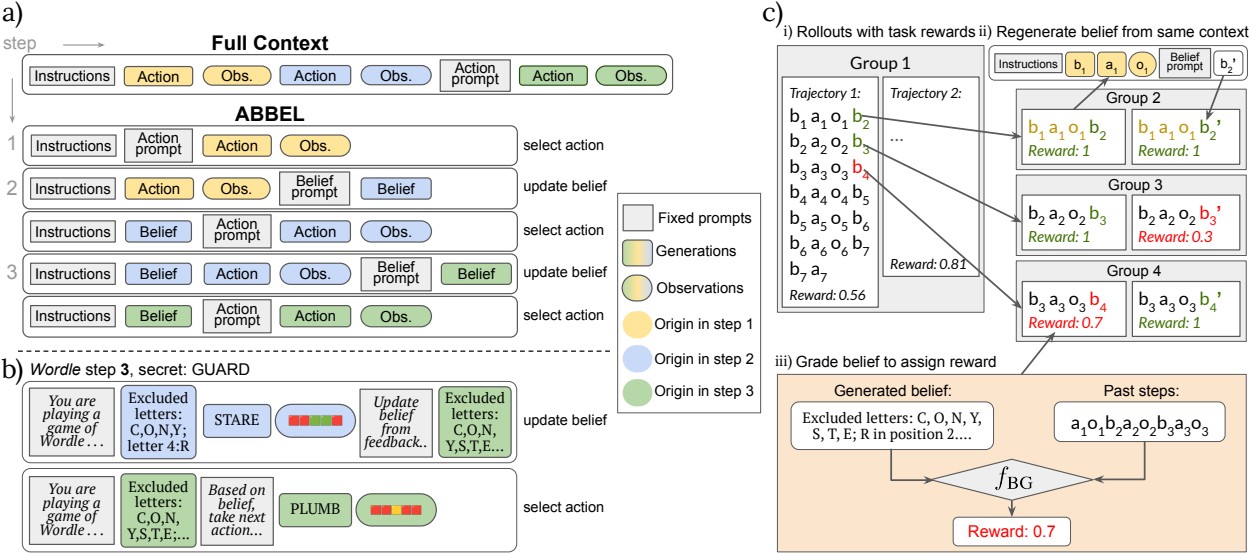

Figure 1: (a) Overview of changes in model context through the typical paradigm (Full Context), which keeps all steps in context, and ABBEL, which alternates between updating beliefs on new observations, and selecting actions based on the last belief. (b) An example ABBEL step in *Wordle*, a game where the player must identify a secret word. Here, actions are word guesses, and observations provide feedback on each letter (i.e., in the correct or incorrect position, or not in the secret at all). (c) Belief grading overview. Beliefs, actions and observations generated at timestep $t$ are denoted by $b_t$, $a_t$, and $o_t$, respectively. After collecting trajectories from the current policy (i), each belief update step is copied into a new group consisting of the original belief update, and additional posterior beliefs generated by re-prompting the model with the same context (ii), which are each assigned rewards by a belief grader (iii). The policy is updated with GRPO using both trajectory and belief groups.

to generate effective belief states based on the task instructions, while RLFT subsequently optimizes belief generation for any given environment.

We first diagnose key failure modes of multi-step recursive summarization by applying ABBEL's rollout framework to out-of-the-box frontier models across five diverse environments and analyzing the resulting belief states. For each model we identify settings where performance is worse than providing the same model with access to the entire interaction history, and trace the causes to belief update errors or omissions that are propagated to future steps. In addition, while we find that belief lengths generally grow much more slowly than interaction histories, we discover settings where memory is used inefficiently due to excess detail or verbosity in the belief states. End-to-end RLFT alone does not penalize this inefficiency, as task outcome-based rewards are agnostic to the lengths of the summaries.

To generate more accurate and parsimonious belief states, we introduce general methods to directly supervise them during RLFT. We propose *belief grading* (Fig. 1c), which creates auxiliary tasks to reinforce the information contents of the beliefs, and *peak belief penalties* (PBP), which incentivizes models to reduce the lengths of belief states with the greatest memory footprints. Belief grading is a flexible method for improving task performance, which allows the developer to leverage domain-specific knowledge with the choice of grading function (Fig. 1c iii). We show that using a general grader based on how well past information can be reconstructed from the belief state doubles sample efficiency and cuts ABBEL's performance gap with full context models from 21% to 8% in a collaborative coding environment. In a numerical deduction game for which a domain-specific grader is available, we demonstrate that belief grading fully closes the gap and further reduces memory usage. We also experiment on a multi-objective QA setting proposed by Zhou et al. (2026) with significantly longer observations and full context lengths of up to 10k tokens. Even without grading, we find training with ABBEL's disentangled beliefs performs on par with full context models, and outperforms MEM1 while using a similar amount of memory. We test the ability of peak belief penalties to further

improve memory efficiency, showing that it reduces memory usage with minimal impact on performance such that ABBEL-PBP uses 67% as much memory as MEM1 while maintaining 40% higher performance. Our results demonstrate that ABBEL is a effective, interpretable and versatile approach to multi-step context management, enabling new forms of supervision and controllability during training.

## 2 Technical Overview

**Problem Setup.** We model each environment as a Partially Observable Markov Decision Process, using *Wordle* as a grounding example. In *Wordle*, the objective is to identify a secret 5-letter word in fewer than 7 steps by guessing a word at each step. Each *task* corresponds to a randomly sampled hidden initial state $s_0$, e.g., (secret:GUARD, step:0). At each step the agent selects an action $a_t$, e.g., a 5-letter word. The hidden state $s_{t+1}$ is updated based on $s_t$ and $a_t$, which in *Wordle* simply increments the step counter. The agent receives reward $r_t$ and observation $o_t$, both conditioned on $a_t$ and $s_t$, e.g., $r_t = 1$ if $a_t = $ GUARD and step $< 7$ otherwise $r_t = 0$, and $o_t$ is feedback on each letter in $a_t$. Performance in an environment is measured by the expected performance across tasks, e.g., over games with different secret words.

**Rollout Frameworks.** We use LLMs to implement context-conditioned policies $a_t \sim \pi(\cdot \mid c_t)$. See Appendix A for the full details and prompts.

**FULL CTX**: Under the typical multi-step paradigm, the agent generates actions conditioned on the environment instructions $p_I$ (e.g., how to play *Wordle*), the full interaction history, $h_t = \langle a_1, o_1, a_2, o_2, \ldots, a_{t-1}, o_{t-1}\rangle$, and action prompt $p_a$ (e.g., "make your next guess of the secret word"), i.e.,

$$a_t \sim \pi(\cdot \mid p_I, h_t, p_a), \tag{1}$$

resulting in a new observation $o_t$ that is directly appended to $h_t$ for the next step.

**ABBEL**: Under ABBEL, the agent generates actions conditioned on a belief state $b_t$ instead of $h_t$, thus reducing the context length when $b_t$ is significantly shorter. The agent is called twice at each step $t$. First, it is called to update its belief state. Specifically, $\pi$ is prompted with $p_I$ and the prior belief, action, observation, and a domain-general belief prompt $p_b$, to generate a posterior belief (*update belief* in Fig. 1):

$$b_t \sim \pi(\cdot \mid p_I, b_{t-1}, a_{t-1}, o_{t-1}, p_b). \tag{2}$$

ABBEL may use chain-of-thought (CoT) prompting for belief generation, but the belief state is generated within tags so that it can extracted, for example:

---

**Context for Belief Update Step**

```
environment_instructions
```
*Your current belief state:* `<belief>...</belief>`
*Your last action:* `<action>...</action>`
*Environment feedback:* `<environment>...</environment>`
*Now update your belief state to include all important new information you have gathered. Do not say anything about future actions. Think step by step and then output your new belief state inside* `<belief>`
`... </belief>`*, e.g.,* `<think>`*Any thinking*`</think>``<belief>`*your new beliefs*`</belief>`*.*

---

Second, the agent is called to choose an action with action prompt $p_a$ and the posterior belief $b_t$ (*select action* in Fig. 1):

$$a_t \sim \pi(\cdot \mid p_I, b_t, p_a), \tag{3}$$

resulting in observation $o_t$, which is subsequently incorporated into belief state $b_{t+1}$ in the next belief update step. While other self-summarization approaches call the agent only once per step, entangling belief updating with action selection reasoning (Zhou et al., 2026), our formulation cleanly separates the belief state, making it possible to study and shape its contents.

**FC BELIEFS**: To disentangle the effect of belief generation from the context bottleneck, we also study belief prompting while keeping the full interaction history in the context. At each step, we first prompt

for belief generation and then prompt for action selection, but both are conditioned on $h_t$ which now also includes beliefs: $h_t = \langle a_1, o_1, b_2, a_2, o_2, \ldots, b_{t-1}, a_{t-1}, o_{t-1}, b_t \rangle$.

## 3 Related Work

**Long context management.** Several recent systems have developed practical solutions for managing long contexts. Context compression methods generate dense representations that, while computationally efficient, sacrifice human-understandability (Chevalier et al., 2023; Jiang et al., 2024). Wang et al. (2025b), Örwall (2024) and Starace et al. (2025) hand-design summarization prompts and pruning strategies specific to their target environments, which requires expert human knowledge of what information must be maintained for each task rather than allowing the agent to learn what to remember as part of its decision-making strategy. Another popular family of methods process long contexts into an external memory store for the agents to query (Packer et al., 2024; Xu et al., 2025; Wang et al., 2023; Zhang et al., 2026); these systems of non-parametric external memory are complimentary to context summarization approaches which keep memory in context, and involve different trade-offs, e.g., they introduce additional engineering overhead and are difficult to optimize jointly with the agent's policy. Wang et al. (2025a) and Yu et al. (2026) use recursive self-summarization to condense long documents, but they are limited to the single-step setting for summarizing static contexts. Sequential decision-making tasks introduce challenges these methods are not equipped for: maintaining and reasoning over knowledge not only for taking immediate action, but also for soliciting additional useful information.

**Information tracking for sequential decision-making.** Various works have studied representations of interaction history for multi-step partially-observable environments. ABBEL is inspired by recursive Bayesian estimation, a foundational framework for integrating information gathered over time by iteratively updating beliefs over the unknown environment (Kalman, 1960; Ho & Lee, 1964). Classical results on sequential decision-making show that this posterior belief state is a minimal sufficient statistic for optimal control (Åström, 1965), so compressing an interaction history into such a belief state could, in principle, limit the growth of the context length without harming performance. However, these early works use exact prior and conditional probability distributions to compute belief updates, which are generally unavailable or computationally intractable to work with in realistic settings. Subsequent work has therefore explored learning latent representations of interaction history from scratch using recurrent neural networks (Hochreiter, 1997; Chung et al., 2014), but these models are challenging to optimize (Pascanu et al., 2013), and the resulting latent representations are difficult to interpret. Hard-coded summary statistics of past observations have proven effective for bandit problems (Krishnamurthy et al., 2024; Nie et al., 2025), but lack the flexibility needed for more complex environments.

**LLM agents updating natural-language memories.** LLMs can represent knowledge flexibly without sacrificing interpretability by using natural-language, and large-scale pretraining initializes LLMs with effective priors for what to remember for any given task. Arumugam & Griffiths (2026) show that frontier models can effectively update and act on natural-language descriptions of belief states instead of interaction histories, but they hand-craft prior beliefs for each environment, and use a suboptimal predefined algorithm to select actions rather than learning to explore optimally from beliefs. Paischer et al. (2023) train RL agents to act on LLM-generated summaries of past events, but they rely on frozen pretrained models for the summarization, which limits performance. Zhou et al. (2026) recently train LLMs end-to-end in multi-step environments to both generate and act on natural-language summaries, but they use the entire action reasoning trace as the memory rather than generating separate beliefs from which to select actions. Entangling memory with reasoning retains irrelevant information, reduces interpretability, and does not allow for direct supervision of summary length and contents.

## 4 Belief State Analysis of Frontier Model Recursive Summarization

We first measure performance and identify failure modes in recursive summarization by applying ABBEL's rollout framework to out-of-the-box frontier models and analyzing the resulting belief states.

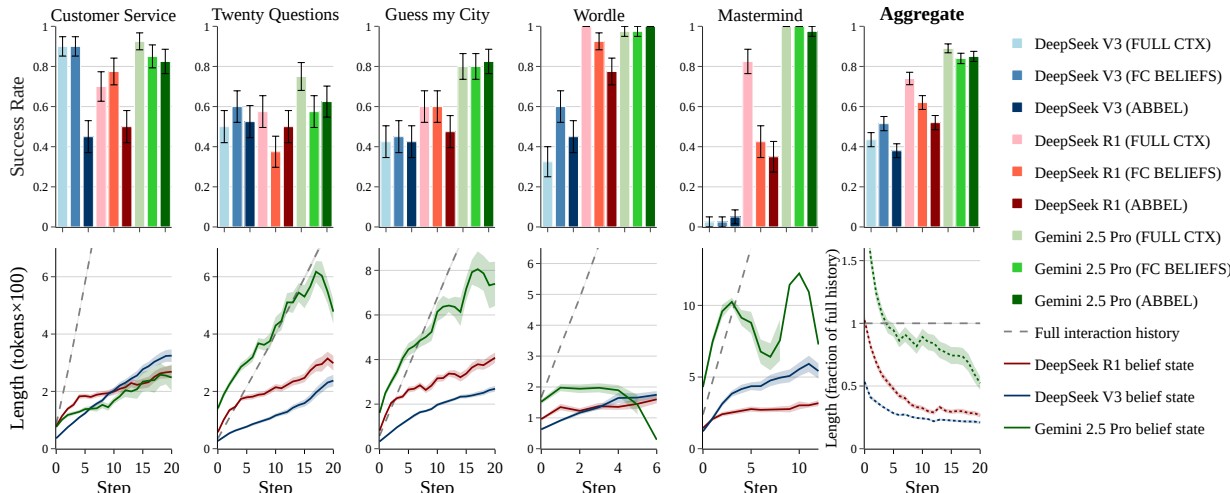

Figure 2: Zero-shot frontier model behavior across interaction frameworks and environments. The **Aggregate** column shows each metric computed from data pooled across environments. Top: average success rates. Models generally perform worse conditioned on summaries alone (ABBEL, darkest shades) than the full interaction history (FULL CTX, lightest shades), though the gap is smallest for Gemini 2.5 Pro. Bottom: context compression, visualized by the lengths of ABBEL belief states compared to full interaction histories after each interaction step. Belief lengths grow much more slowly than full histories, with the exception of Gemini 2.5 Pro. No model is consistently both performant and memory-efficient under recursive summarization.

## 4.1 Experimental Setup

**Environments.** We evaluate across five multi-step environments from Tajwar et al. (2025) spanning various levels of reasoning complexity and structure.[1] *Wordle* and *Mastermind* demand complex reasoning using highly structured feedback on a secret word or 4-digit code. *Mastermind* has the same rules as *Wordle* (described in Section 2), except feedback only reveals two pieces of information: the number of guessed digits in the correct position, and the number of guessed digits in the code but in the wrong position. Due to the complexity of the reasoning in these two environments, we follow Tajwar et al. (2025) in using CoT prompting for action selection. The remaining three environments are less structured, and are simulated using GPT-4o-mini. *Twenty Questions* and *Guess My City* involve iteratively narrowing down a search space of topics or cities by asking a sequence of questions. *Customer Service* is the least structured setting, where actions and observations may span multiple sentences: the goal is to correctly diagnose a customer's problem in scenarios ranging from electronic device issues to automobile maintenance, by issuing troubleshooting instructions and reasoning over the customer's responses.

**Metrics.** All environments return a binary success indicator; we sample 40 task instances from each and report the mean and standard error of the mean (Success Rate). To measure the degree to which the belief states compress the interaction history, we compare their lengths at each step.[2]

**Models and Frameworks.** We evaluate frontier reasoning models Gemini 2.5 Pro and DeepSeek R1, and a similar non-reasoning model DeepSeek V3 to study any differences in bottleneck behavior between reasoning and non-reasoning models. For each, we compare their behavior through ABBEL with the standard full context setting (FULL CTX), and full context with belief prompting (FC BELIEFS) as described in Section 2.

## 4.2 Results

**Task Performance.** We first analyze the success rates under each framework (Fig. 2, top), confirming that ABBEL replicates the performance gap reported by Cassano & Rush (2026) between recursive summarization

---

[1]Table 3 summarizes key characteristics of each environment.
[2]We could not directly measure overall memory usage because some model APIs did not provide raw reasoning token counts.

models and models with full context access (FULL CTX) across many settings: performance decreases by up to 50%, 58%, and 17% for DeepSeek V3, R1 and Gemini 2.5 Pro, respectively. In the settings with a significant gap, ABBEL generally also underperforms agents conditioned on both beliefs and the full interaction history (FC BELIEFS), providing further evidence that the generated beliefs may be excluding valuable information from the history. By inspecting the belief states, we find the drop in performance can indeed often be attributed to omissions or errors in the beliefs, which, once introduced, are propagated to future timesteps. These errors are most common in environments requiring complex reasoning to update beliefs (*Wordle* and *Mastermind*). Models self-correct if they receive contradictory observations, but the wasted turns may be irrecoverable; whereas access to the full history makes it possible to detect errors earlier and reconstruct the correct posterior from all past observations. We find two main causes of belief state errors: incorrectly updating on the new observation due to mistakes in reasoning (e.g., falsely assuming that the secret code cannot contain repeated characters), and hallucinating false memories of past interactions (see Appendix C.3 for an example).

**Belief State Compactness and Interpretability.** We find that in most cases, the belief states are significantly shorter than the length of the full interaction history past the first few steps (Fig. 2, bottom), showing that ABBEL can effectively reduce context lengths. By step 20, the belief lengths are on average 21%, 27% and 52% the length of the full histories for DeepSeek V3, R1 and Gemini 2.5 Pro, respectively. While the history always grows linearly with the number of interaction steps, belief lengths grow more slowly, plateauing or even decreasing in some environments as possibilities were ruled out. However, Gemini 2.5 Pro's belief states in *Twenty Questions* and *Guess My City* are a notable exception, exceeding the full context length without significant performance gains in *Twenty Questions* compared to DeepSeek R1. By inspection we found that all models generate interpretable belief states, which allow us to better understand model behavior: Gemini 2.5 Pro concatenates all answers to its questions verbatim, which explains why the length grows linearly, whereas DeepSeek R1 maintains a compact summary (see Appendix B for examples).

**Reasoning Efficiency.** ABBEL splits reasoning at each interaction step into two LLM calls: first, to update the belief, and second, to reason over the posterior belief to select an action. If belief states are a natural information bottleneck in multi-step reasoning, then ABBEL should use fewer reasoning tokens during action selection, and the total number of reasoning tokens generated at each step would increase by a factor of less than 2. Indeed, controlling for performance, we find that conditioning on belief states generated by ABBEL significantly reduces action selection reasoning length in most environments, using on average 62% and 19% fewer tokens than FULL CTX with DeepSeek R1 and Gemini 2.5 Pro, respectively (Fig. 5, middle). Inspecting reasoning traces, we observe FULL CTX models often already generate a description of beliefs as the first step of reasoning, and so conditioning on an explicit belief state reduces this reasoning. Thus, compared to FULL CTX, DeepSeek R1 surprisingly generates 24% *fewer* total reasoning tokens per step under ABBEL, and Gemini 2.5 Pro only generates 48% more (Fig. 5, bottom). We also find models often use even less reasoning under ABBEL than FC BELIEFS. Inspecting the traces show that access to full histories results in regenerating beliefs from scratch (see Appendix C.2 for examples). Thus, using belief states as bottlenecks in reasoning provides an additional benefit of preventing unnecessary extra reasoning over interaction histories when beliefs are sufficient. See Appendix C.1 for more details.

## 5 Improving Belief Generation with Reinforcement Learning

By inspecting the belief states generated under ABBEL's rollout framework, we found in Section 4 that recursive summarization suffers from reduced performance or memory efficiency due to omitting information, introducing errors, or retaining extraneous information. We address these limitations by directly reinforcing the length and information contents of each belief state.

We use reinforcement learning (RL), as supervised fine-tuning on expert demonstrations of belief updating are generally not available, and RL has been shown to improve general abilities across distribution shifts compared to SFT alone (Nie et al., 2025; Kirk et al., 2024; Tajwar et al., 2025). We use GRPO as our RL algorithm, which has been shown to be more stable and effective for training LLM reasoning than PPO (Shao et al., 2024). Trajectory rollouts are collected through the multi-step frameworks described in Section 2, with CoT prompting for both belief generation and action selection. We collect multiple rollouts on each task

instance to form groups, each trajectory is assigned a scalar reward based on the outcome, e.g., how many steps it took to generate the correct answer (Fig. 1c i), and then advantages are calculated by normalizing the rewards within each group. Although the tokens generated under ABBEL are conditioned on disjoint contexts across interaction steps, all tokens generated in the same trajectory are assigned the same advantage for training.

## 5.1 Belief Grading

Prior work has found that learning sophisticated reasoning with only sparse outcome-based rewards can be challenging (Zhang & Zuo, 2025). In environments requiring complex belief update reasoning such as *Wordle* and *Mastermind*, we hypothesize that this would result in difficulty learning to generate accurate belief states. A well-established approach for enriching the learning signal in sparse reward settings is the use of auxiliary tasks, i.e., creating additional tasks to train the agent on such that they learn useful behaviors for eventually maximizing the real rewards (Jaderberg et al., 2017). Thus, we propose *belief grading*, a novel auxiliary task-based approach for supervising belief generation, inspired by recent work evaluating context summarization prompts by grading the quality of the summaries (Wang et al., 2025b).

An overview of belief grading is presented in Fig. 1c and Algorithm 1. As in standard multi-step RL, during training we roll out trajectories in the environment comprising observations, actions and beliefs sampled from $\pi_\theta$. To support belief grading, we also sample additional belief 'trajectories' consisting of single belief update generations to which we assign the grading rewards. For each belief in the full environment trajectory we prompt the agent to re-generate beliefs from the same context to create a group for GRPO (shown for group size 2 in Fig. 1c ii). We then use a belief grader function $f_{BG}$, which may be conditioned on the full interaction history, and potentially other domain or task-specific information, to assign scalar rewards to each belief (Fig. 1c iii). This provides a learning signal whenever the beliefs in a group receive different grades. If a belief in a trajectory receives a sufficiently low grade, we do not grade subsequent beliefs in that trajectory, to avoid penalizing posterior beliefs that are only poor because they are conditioned on incorrect prior beliefs. To avoid over-optimizing for the auxiliary rewards, we do not rescale the belief grading advantages by their standard deviation. Finally, the advantages from the belief grading and the task trajectory groups are summed and the policy gradient step is applied.

---

**Algorithm 1** Belief Grading with GRPO (presented with group size 2 for simplicity).

---

**Require:** Environment instructions $p_I$; belief generation prompt $p_b$.
**Require:** ABBEL policy model $\pi_\theta$; batch of trajectories $\mathcal{D} = \{\tau_i\}_{i=1}^N$ rolled out by $\pi_\theta$.
  all_belief_groups ← [ ]
  **for** $\tau$ in $\mathcal{D}$ **do**
    **for** $t$, step  in enumerate($\tau$) **do**        ▷ Create a new group for each belief update step in $\tau$.
      $b_t, a_t, o_t, b_{t+1}$ ← step
      belief_context ← $p_I, b_t, a_t, o_t, p_b$
      $b'_{t+1} \sim \pi_\theta(\cdot|\text{belief\_context})$       ▷ Generate another belief at this step from the same context.
      $r \leftarrow f_{BG}(b_{t+1}, \tau, t)$               ▷ Grade each belief.
      $r' \leftarrow f_{BG}(b'_{t+1}, \tau, t)$
      step_t_belief_group ← [(belief_context, $b_{t+1}, r$), (belief_context, $b'_{t+1}, r'$)]
      all_belief_groups.append(step_t_belief_group)
    **end for**
  **end for**
  Update $\pi_\theta$ with GRPO over all rollout and belief groups.

---

**Belief Grader Functions.** Different grader functions may be used for different environments, providing a flexible way to leverage domain knowledge, as shown in Section 6.1. In the absence of domain-specific knowledge, the grader may be based on the ability to reconstruct information from the interaction history using the belief state. Specifically, inspired by variational auto-encoders (Kingma & Welling, 2013), we propose to grade beliefs by treating the language model $\pi_\theta$ as both encoder and decoder of information from the trajectory, and the belief states as the codes. We can then compute the grade of posterior belief $b_{t+1}$

as the log probability under $\pi_\theta$ of tokens from previous steps (which could be observations, actions, and/or prior beliefs), conditioned on $b_{t+1}$. See Section 6.2 for a concrete instantiation of this idea.

## 5.2 Peak Belief Penalties

Outcome-based rewards only incentivize task performance, not memory efficiency. Thus, in settings where models are predisposed to generate bloated belief states (e.g., Gemini 2.5 Pro in *Twenty Questions*), incorporating a penalty based on belief lengths into the training objective could lead to large efficiency gains. Additionally, in settings where memory is especially limited, such a penalty could allow for efficient trade-offs between memory and performance. Because ABBEL's belief states are separated from the reasoning, a belief length penalty can encourage more concise beliefs without degrading reasoning capabilities.

The penalty for a trajectory is based on the token count of the longest (i.e., peak) belief state in the trajectory, because memory usage is determined by the maximum context lengths. To avoid belief over-compression at the expense of task performance, we find two additional design choices are important. First, we subtract a minimum token threshold $T_{\min}$ from the peak belief length corresponding to an acceptable memory usage, such that the penalty is proportional to the number of tokens exceeding $T_{\min}$. Second, we normalize the trajectory outcome-based advantages before adding the un-normalized penalties, so that their impact relative to the outcome rewards decreases as beliefs get shorter, which was shown to be helpful in prior token efficiency work (Arora & Zanette, 2026). The penalties are finally scaled by a constant factor $k_{\mathrm{PBP}}$ which is chosen by the developer to control the degree to which memory reduction should be prioritized over task outcomes. Thus the penalty for each trajectory is calculated as follows:

$$\mathrm{PBP}(a_1, o_1, b_2, a_2, ..., b_n, a_n) = -k_{\mathrm{PBP}} \max(0, \max_t |b_t| - T_{\min}). \tag{4}$$

# 6 Reinforcement Learning Experiments

We evaluate belief grading and peak belief penalties in different settings where each is expected to be most beneficial. We demonstrate the ability of belief grading to improve performance in a numerical deduction game and a conversational collaborative coding setting, which require complex reasoning for updating beliefs. We study the impact of peak belief penalties on memory efficiency in a more memory-intensive multi-objective QA domain, which involves much longer observations and horizons. Across all environments, we measure memory usage with the *Peak Tokens* metric proposed by Zhou et al. (2026), which is the maximum sequence length (input and output, excluding the system prompt) over all steps in each trajectory. We train Qwen2.5-7B-Instruct in all settings. See Appendix D for more details.

## 6.1 Belief Grading with Domain Knowledge

We first demonstrate the efficacy of belief grading in an environment requiring complex belief update reasoning, for which we could leverage domain knowledge to design a domain-specific grader function.

**Environment and Metrics.** *Combination Lock* is a 3-character version of *Wordle* proposed by Arumugam & Griffiths (2026). We train with a vocabulary of 10 digits and a 12-step horizon, and to test generalization we evaluate on a disjoint vocabulary of 16 letters and a 16-step horizon. Each episode ends with reward $(H + 1 - \text{steps to find code})/H$ if the code was identified, and $-1$ otherwise. We report the mean *Regret*, which is the number of incorrect attempts made in an episode.

**Experimental Setup.** We train ABBEL with belief grading using a domain-specific grader (ABBEL-dom-BG), as well as an ablation of grading by training only with outcome reward (ABBEL), and the full-context settings FC BELIEFS and FULL CTX described in Section 2. For each framework we also measure zero-shot (Zero) performance to evaluate the effect of RLFT. We finally train Qwen2.5-14B-Instruct to study the scalability of our approach to larger models. We train Qwen2.5-7B and Qwen2.5-14B for 140 and 100 steps, respectively, using three random seeds for each setting.

**Belief Grader Function**. In *Combination Lock*, the set of possible numbers at each position is a highly informative statistic computable from the interaction history. To grade each generated belief state $b_t$, we

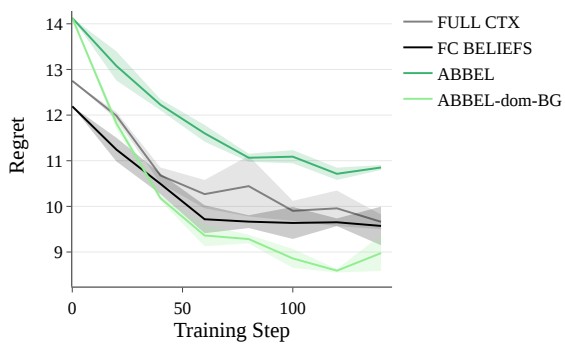
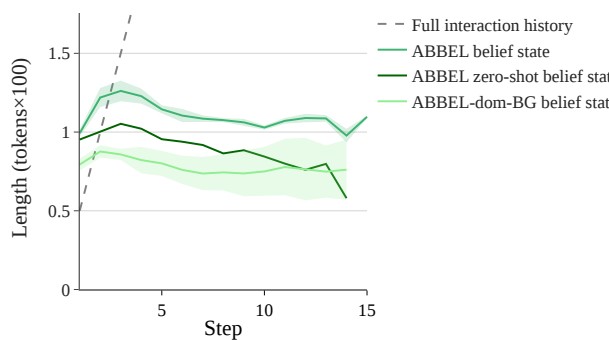

(a) Evaluation Regret (lower is better) over training, showing that fine-tuning ABBEL with belief grading is effective.

(b) Average lengths of belief states compared to the full interaction history after each interaction step.

Figure 3: Results of training Qwen2.5-7B in *Combination Lock*. Belief grading with a domain-specific grader (dom-BG) allows ABBEL to surpass the performance of models trained with full context access (FULL CTX, FC BELIEFS) and generate even more compact belief states. Shaded regions indicate SEM over 3 seeds.

Table 1: Model comparison in *Combination Lock*. Arrows indicate desired directions. We report the mean and SEM over 3 training seeds for all trained models, and over the test set for zero-shot models.

| Framework | Training | Qwen2.5-7B | | Qwen2.5-14B | |
|---|---|---|---|---|---|
| | | Regret ↓ | Peak Tokens×$10^2$ ↓ | Regret ↓ | Peak Tokens×$10^2$ ↓ |
| FULL CTX | Zero-shot | 12.8±0.2 | 23.0±0.4 | 9.5±0.3 | 14.3±0.4 |
| | RLFT | 9.7±0.2 | 20.5±0.1 | **8.3±0.1** | 13.6±0.2 |
| ABBEL | Zero-shot | 14.1±0.2 | 9.4±0.2 | 11.0±0.4 | **7.0±0.1** |
| | RLFT | 10.8±0.1 | 10.5±0.1 | 9.8±0.3 | 9.1±0.7 |
| | **RLFT+dom-BG** | **9.0±0.4** | **8.0±0.2** | **8.3±0.1** | 7.4±0.2 |

computed these sets and compared them to the result of parsing the contents of $b_t$ into the same format[3], generating a grade of 1 when they were identical and 0 otherwise (Algorithm 3). We stop grading each trajectory after the first step with an incorrect belief, to avoid penalizing beliefs that were only incorrect due to propagating errors from the previous step.

**Results.** For the 7B models, training ABBEL with belief grading is highly effective, resulting in ABBEL-dom-BG *outperforming* the FULL CTX agent by 7% (Fig. 3a). The belief states remain concise, and even decrease in length compared to ABBEL zero-shot (Fig. 3b, see Appendix D.1 for examples), resulting in a 60% reduction in memory usage compared to FULL CTX. Without grading, ABBEL's Regret converges above FULL CTX while using more memory than ABBEL-dom-BG (see Table 1). Comparing FULL CTX with FC BELIEFS, we find that belief prompting helps initially, but the advantage disappears over training. All 14B agents started with lower zero-shot Regret and Peak Token usage than their 7B counterparts (see Table 1), but we again find that RLFT with belief grading significantly boosts ABBEL-dom-BG's performance and memory efficiency, reaching FULL CTX Regret with about half the memory usage.

## 6.2 Belief Grading with Information Reconstruction

We demonstrate that belief grading generalizes to a less synthetic setting, using a general grader based on reconstructing past information from the belief state.

**Environment and Metrics.** We evaluate using ColBench, the collaborative back-end programming environment introduced by Zhou et al. (2025), where the agent must communicate with a human user to write a Python function of up to 50 lines. The agent is initially provided with an under-specified high level description and the function signature, and can ask up to 10 questions before finally submitting code. The

---

[3]We used Gemini 3 Flash for parsing, but we note that parsing could be replaced by the use of structured output schemas.

generated code is finally evaluated by 10 hidden unit tests, yielding an outcome reward equal to the fraction of tests passed. We report the mean fraction of passing tests (*Test Pass Rate*), and the fraction of tasks with all 10 tests passing (*Success Rate*). The user is simulated by Gemma 3 27B-it with access to the hidden tests and a reference solution, prompted to behave like a human that needs help.

**Experimental Setup.** We train five seeds each of ABBEL with and without information reconstruction belief grading (rec-BG), and the full-context setting as a measure of best-case performance (FULL CTX), evaluating after 0 (-Zero), 50 and 100 training steps.

**Belief Grader Function.** We grade each generated belief $b_{t+1}$ by how well $\pi_\theta$ can reconstruct the most recent observation $o_t$ given $b_{t+1}$, $b_t$ and action $a_t$, thus rewarding $b_{t+1}$ for integrating information in $o_t$ that is not already in the prior $b_t$. We compute this as the log probability under $\pi_\theta$ of the tokens in the last observation $o_t$, conditioned on the environment instructions $p_I$ and $b_t$, $a_t$, and $b_{t+1}$, i.e.,

$$f_{\mathrm{BG}}(b_{t+1}) = \log \pi_\theta(o_t \mid p_I, b_t, a_t, b_{t+1}). \tag{5}$$

By application of Bayes' rule, we derive that the $b_{t+1}$-dependent component of this is proportional to

$$\log \pi_\theta(b_{t+1} \mid p_I, b_t, a_t, o_t) - \log \pi_\theta(b_{t+1} \mid p_I, b_t, a_t), \tag{6}$$

where the second term encourages $b_{t+1}$ to contain new information relative to $p_I$, $b_t$ and $a_t$, while the first encourages that new information to be explainable by $o_t$. See Algorithm 4 for details.

**Results.** Results are shown in Table 2. We find that reconstruction-based belief grading is effective: though ABBEL-rec-BG does not outperform FULL CTX, it reduces ABBEL's performance gap by about 2x and trains in half the steps (50 vs 100), while remaining highly memory-efficient: its test pass rate is only 7.7% lower than FULL CTX while using 57% less memory. We observe that the grading helps ABBEL learn to add more useful information to its beliefs, as ABBEL-rec-BG's belief states were longer on average (see Appendix D.3 for examples). We observe zero-shot ABBEL and FULL CTX perform similarly due to ABBEL agents having a helpful initial bias to ask for more clarifications before submitting an answer, as we observed in Section 4.2.

Table 2: Model comparison on ColBench. Arrows indicate desired directions. We report the mean and SEM over 5 training seeds for all trained models, and over the test set for Zero-shot models.

| Model | Test Pass Rate ↑ | Success Rate ↑ | Peak Tokens$\times 10^2 \downarrow$ |
|---|---|---|---|
| FULL CTX-Zero | 0.283±0.013 | 0.175±0.012 | 4.594±0.153 |
| FULL CTX | 0.519±0.019 | 0.393±0.021 | 14.078±0.547 |
| ABBEL-Zero | 0.264±0.013 | 0.171±0.012 | **3.295±0.053** |
| ABBEL | 0.455±0.017 | 0.314±0.016 | 4.203±0.367 |
| ABBEL-rec-BG | **0.479±0.009** | **0.356±0.011** | 6.014±0.330 |

### 6.3 Learning with Peak Belief Penalties

Finally, we test the ability of peak belief penalties (PBP) to improve memory efficiency in a setting with much lengthier 300-word observations and extreme horizon generalization (from 2 questions and 6 steps to 16 questions and 20 steps). We do not apply belief grading in this domain, as we find ABBEL already matches the performance of agents with full context access.

**Environment and Metrics.** In the multi-objective QA environment introduced by Zhou et al. (2026) each task requires the agent to answer a list of questions (objectives) by iteratively querying a database before generating a final answer composed of semicolon-delimited answers to each question. Each search query, which consists of keywords or a short phrase, retrieves the first 100 words of the three most relevant documents in the database. During training, each task involves only 2 questions and a horizon of 6 steps, while we evaluate on tasks with up to 16 objectives and 20 steps. We use the *Exact Match Count* (EM) – the number of answers that exactly match the correct answer text – as both the task reward and performance metric. In addition to the Peak Tokens metric, we also measure peak memory state lengths, i.e., the lengths of the longest belief or internal state in each trajectory, to more directly study the effect of peak belief penalties.

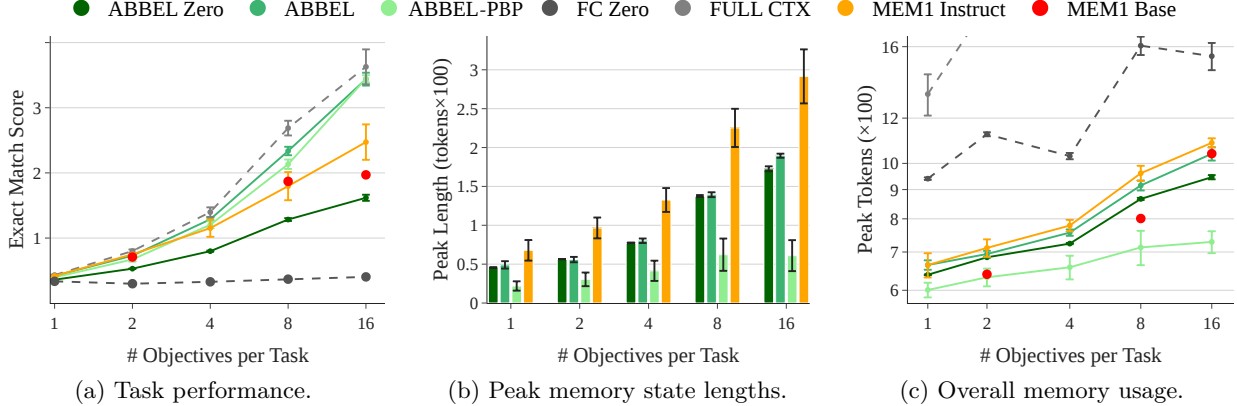

(a) Task performance.     (b) Peak memory state lengths.     (c) Overall memory usage.

Figure 4: Model comparison in multi-objective QA. The mean and SEM over 3 seeds is shown for all trained models, and over the test set for Zero-shot models. ABBEL trained with peak belief penalties (ABBEL-PBP) significantly outperforms prior work (MEM1), while using much less memory. Ablating PBP shows it led to much more concise belief states (b) with minimal impact on performance. See Table 9 for numbers.

**Experimental Setup.** We evaluate ABBEL trained with peak belief penalties (ABBEL-PBP, see Appendix D.2 for details), and ablations with no PBP (ABBEL) and no training (ABBEL Zero). We compare with MEM1 (Zhou et al., 2026), which also uses RL to train LLMs to generate and act on context summaries. In MEM1, however, rather than generating a separate belief state, the entire reasoning trace (defined as the 'internal state') is carried forward as the memory to the next step, so a length penalty could not be applied without also discouraging reasoning. We take the results reported by Zhou et al. (2026) for MEM1 (MEM1 Base, trained with PPO from Qwen2.5-7B-Base). We also re-implement MEM1 by training a Qwen2.5-7B-Instruct model with GRPO under MEM1's prompting and rollout framework (MEM1 Instruct)[4] for a direct comparison with ABBEL. As a measure of best-case performance, we train a Qwen2.5-7B-Instruct model in the full-context setting (FULL CTX) and also evaluate its zero-shot performance (FC Zero). We train three random seeds for each model.

**Results.** Settings with more objectives amplify differences between models, as they require more steps of information gathering. ABBEL-PBP achieves significantly higher performance than all MEM1 models for greater than 4 objectives (Fig. 4a) while using the least memory (Fig. 4c), reaching a 40% performance gain with 33% less memory at 16 objectives. ABBEL-PBP even achieves close to FULL CTX performance, with no significant gap at 16 objectives while using 13x less memory, and also surpasses a representative external memory module approach (Xu et al., 2025) on both metrics[5] (see Table 9).

Inspecting the belief states, we find that they remain interpretable, summarizing what is known so far about the answers to the questions, whereas MEM1's internal states are significantly longer (Fig. 4b), containing reasoning for drawing conclusions from previous search results (see Appendix D.2 for examples). Though MEM1 makes one LLM call per step compared to two for ABBEL, for 16 objectives we found it did not generate significantly fewer total tokens per step than ABBEL-PBP, and only used 23% fewer total input tokens, which are cheaper due to parallelizability (see Table 8). Comparing ABBEL-PBP to ABBEL shows that PBP enables large gains in memory efficiency: it significantly cuts memory usage by reducing the peak belief state lengths by more than 2x (Fig. 4b), while task performance is on par with or close to ABBEL across all objective counts. ABBEL's belief states are still shorter than MEM1's internal states because they do not contain reasoning; the more concise memories may have contributed to ABBEL's superior performance, by being easier to reason over.

---

[4]In our experiments we found training from Qwen2.5-7B-Instruct outperformed training from the base model.

[5]Results taken from Zhou et al. (2026) for comparison.

## 7 Discussion

We introduce ABBEL, a framework for training LLMs to recursively update beliefs for interpretable, efficient multi-step context management. We first identify key failure modes of recursive summarization by analyzing the belief states generated by frontier models under ABBEL's rollout framework. We find that models underperform full context agents due to omitting information or introducing errors, and also discover settings where models retain significant amounts of extraneous information. We target these limitations with methods to supervise belief generation during RLFT: we propose *belief grading*, a flexible auxiliary task method for supervising the generation of higher quality beliefs, and *peak belief penalties* to reduce overlong beliefs without degrading reasoning. We show that belief grading is a powerful and versatile technique: an information reconstruction-based grader significantly improves performance and sample efficiency, and a domain-specific grader fully closes the gap with full context models. We finally demonstrate peak belief penalties enable large gains in memory efficiency, allowing ABBEL to outperform prior work by a wide margin while using significantly less memory.

Studying ABBEL in more realistic settings is an important future direction. For instance, ABBEL updates beliefs at every step, whereas in practice summaries may be updated less frequently to reduce inference costs and information loss (Cassano & Rush, 2026). Additionally, although we show ABBEL generalizes beyond its training horizon, we did not evaluate in settings with horizons greater than 20 steps. Several scientific questions also remain open. In particular, we did not examine in detail how supervising belief generation affects action selection and reasoning behavior beyond high level performance and memory usage metrics–it is possible that training separate models for belief generation and action selection may further improve performance by reducing task interference (Li et al., 2026). More broadly, while we only investigate the potential for supervising belief updates, ABBEL also enables other new forms of supervision such as guiding information-seeking actions by their effects on beliefs. Exploring these directions remains a promising avenue for future research.

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

**LLM Usage in paper writing.**

LLM tools were used minimally for finding related work, polishing writing, e.g., rephrasing sentences to flow more naturally, and editing code to reformat figures.

# A   Belief Bottleneck Rollout

See Algorithm 2 for the rollout algorithm. Below, we show one sample and associated prompts for the game *Wordle* with Deepseek R1 (illustrated in Fig. 1).

---

**Algorithm 2** Belief Bottleneck Rollout

---

**Require:** Instructions $p_I$; horizon $H \in \mathbb{N}$; step function $T : \mathcal{S} \times \mathcal{A} \to \mathcal{S} \times \mathcal{O}$; initial state $s_0$.
**Require:** Belief update prompt $p_b$; action selection prompt $p_a$; policy $\pi$.

   $t \leftarrow 0$
   $s \leftarrow s_0$
   $b \leftarrow$ "This is the start of the game. No beliefs right now."
   **while** $t \leq H$ **do**
      $a \sim \pi(\cdot | p_I, b, p_a)$                                          $\triangleright$ Action selection
      $s, o \leftarrow T(s, a)$                      $\triangleright$ Environment step: get new state and observation.
      $t \leftarrow t + 1$
      $b \sim \pi(\cdot | p_I, b, a, o, p_b)$                                 $\triangleright$ Belief update.
   **end while**

---

### Game State

**Secret:** GUARD
**Step:** 2 out of 6
**Goal Reached:** No

---

### Instructions

You are playing a game of Wordle.
Format your response in the following way: `<think>` Any step-by-step, short and concise thinking to strategically determine the next guess for the secret word `</think>` `<Answer>` your guess of what the word should be `</Answer>`
The game begins now, please make your first guess about the secret five-letter word!

---

### Current Belief State

Excluded letters: C, O, N, Y
Included letters: R (must be in position 1, 3, 4, or 5)
Target word contains R and excludes C, O, N, Y.

---

### Action Prompt

This is the game information:
`game_instructions`
You are currently taking your attempt `step` out of `horizon` attempts. Look at the current belief state and history and give an answer based on it.
Give an answer that leads to optimal exploration and do not be greedy unless it is the last attempt.
Try to maximize the amount of information you have so that you can solve the task correctly.
Belief: `curr_belief`
Please format your response as: `<Answer>`your answer in the correct format mentioned`</Answer>`

---

### Action and Observation

**Action:** STARE
**Observation:**
First letter, s, is not in the target word
Second letter, t, is not in the target word
Third letter, a, is correct and in the correct position in the target word
Fourth letter, r, is correct and in the correct position in the target word

---

Table 3: Characteristics of frontier model evaluation environments. Information structure refers to how structured the environment observations are.

| Environment | Horizon | Complex Reasoning | Information Structure | Answer Space Size |
|---|---|---|---|---|
| Customer Service | 20 | No | Low | $\sim 10^2$ (faulty parts) |
| Twenty Questions | 20 | No | Medium | $\sim 10^3$ (e.g., animals) |
| Guess My City | 20 | No | Medium | $\sim 10^3$ (all cities) |
| Wordle | 6 | Yes | High | 2315 (5-letter words) |
| Mastermind | 12 | Yes | High | $10^4$ (4-digit numbers) |

---

Fifth letter, e, is not in the target word

---

**Belief Update Prompt**

This is the game information:
`game_instructions`
Look at the current belief and the agent's action and environment response on that belief. You have to update the current belief based on the action and response, while maintaining important information about the game state needed to take optimal future actions.
Current belief: `curr_belief`
Agent's action: `action`
Environment's response: `observation`
Output the updated belief state inside `<BELIEF>` and `</BELIEF>` tags.
Understand that only the generated belief is fed to the agent to pick the next action, not the history, so be sure to include all necessary information.

---

**Updated Belief State**

Excluded letters: C, O, N, Y, S, T, E
Included letters: A (position 3), R (position 4)
Target word contains A and R, excludes the listed letters, and has A in position 3 and R in position 4.

---

## B  Frontier Model-Generated Belief State Examples

We compare examples of the belief states generated by DeepSeek R1 and Gemini 2.5 Pro operating under ABBEL. Each are at step 15 of the *Twenty Questions* game, where the answer is "waistcoat".

---

**DeepSeek R1**

The garment is a standalone structured, strapless, sleeveless formal bridal piece, not layered over other garments (confirmed). Excludes outerwear (boleros, jackets), full gowns, corsets, bustiers, and undergarments. Focus narrows to standalone bridal separates with stiffened fabrics (taffeta, satin) or layered construction for structural elegance, lacking boning. Key possibilities: embellished strapless bridal tops (beaded/embroidered), corsage-inspired designs with tailored support, or reinforced bridal camisoles/decorative tulle overlays. Must function independently (paired with skirts) without waist-flared elements or basque terminology. Core attributes: bridal formality, structural integrity via tailoring or stiff fabrics, no corsetry associations.

**Gemini 2.5 Pro**

I am playing a game of 20 Questions to identify a piece of clothing. I have now used 15 out of 20 questions.

Here's what I know so far:

- **Fact 1:** The item is worn on the upper body.

- **Fact 2:** It is NOT typically worn as outerwear.

- **Fact 3:** It typically has a collar.

- **Fact 4:** It typically has buttons all the way down the front.

- **Fact 5:** It is typically worn by men.

- **Fact 6:** It is typically worn for formal or business occasions.

- **Fact 7:** It has short sleeves.

- **Fact 8:** It features decorative vertical pleats or embroidery on the front.

- **Fact 9:** It is NOT a Guayabera shirt.

- **Fact 10:** It is NOT considered a form of national dress for a specific country.

- **Fact 11:** It is specifically worn with a tuxedo.

- **Fact 12:** It is NOT called a "short-sleeved tuxedo shirt".

- **Fact 13:** Its decorative front does NOT feature ruffles.

- **Fact 14:** Its decorative front does NOT consist of a distinct, separate panel of fabric (a "bib").

- **Fact 15:** The decorative vertical pleats on the front are NOT known as "pin-tucks".

Based on the cumulative information, my confidence has been updated.

- **Analysis:** The "No" answer to my last question, "Are the decorative vertical pleats on the front known as 'pin-tucks'?" is a major setback. My leading hypothesis of a "pleated-front shirt" or "pin-tuck shirt" has been invalidated. This means that if the decorative elements are pleats, they are a very obscure type.

- **Re-evaluation:** I must now reconsider the other half of Fact 8: "... pleats **or** embroidery". Having exhausted the most common types of pleated fronts (ruffles, bib, pin-tucks), it is highly likely that the decorative elements are not pleats at all, but rather a form of embroidery. The object remains a very unusual item: a short-sleeved formal shirt worn with a tuxedo, featuring vertical decorations that are not ruffles, not a bib, and not pin-tucks.

- **Current Prime Candidate:** There is no longer a single prime candidate for the item's name. The identity is very mysterious. The immediate goal is to confirm the nature of the decoration. My new leading hypothesis is that the item is a shirt with vertical **embroidery** on the front.

My confidence in any specific name is now low. However, I am confident that the next step must be to pivot away from pleats and investigate the "embroidery" possibility directly.

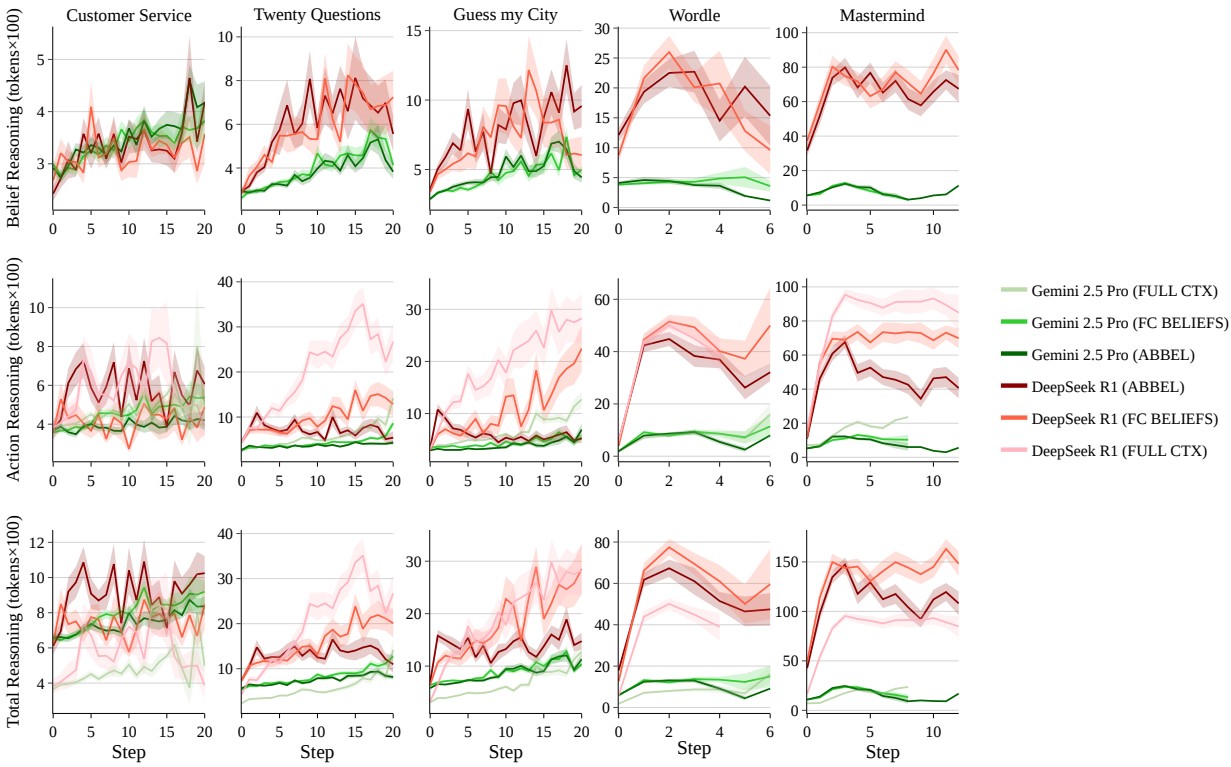

Figure 5: Reasoning length for belief generation (top), action selection (middle), and the total reasoning length at each step, summing the belief and action selection reasoning lengths (bottom). Gemini 2.5 Pro has no data at higher steps when all episodes end early. DeepSeek V3 not shown because it is not a reasoning model.

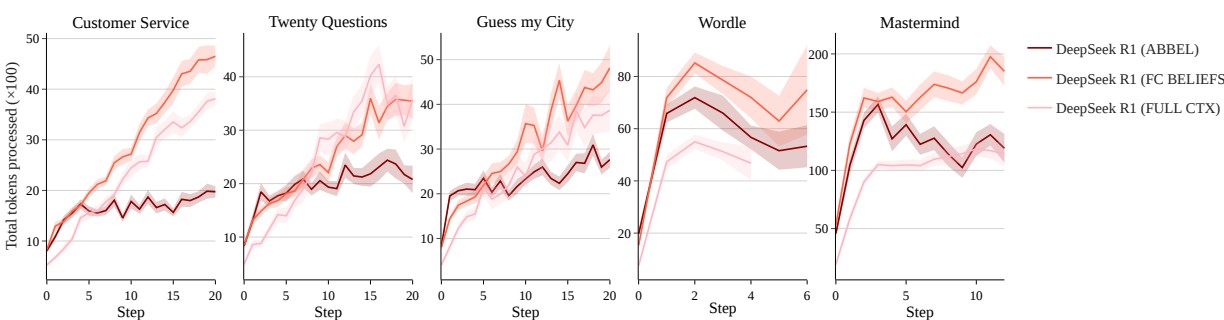

Figure 6: The total number of tokens processed at each step, including both input (i.e., the context) and output (i.e., reasoning, actions and belief states). This remains almost constant for ABBEL, while in many environments it increases nearly linearly for the other frameworks.

## C  Frontier Model Reasoning Analysis

### C.1  Reasoning Lengths

Figure 5 shows the average length of reasoning used for belief generation, action selection, and the total reasoning at each step for DeepSeek-R1 and Gemini-2.5-Pro. Only reasoning summaries, rather than full reasoning traces, were available for Gemini-2.5-Pro. We assume that lengths of reasoning summaries correlate with total reasoning length.

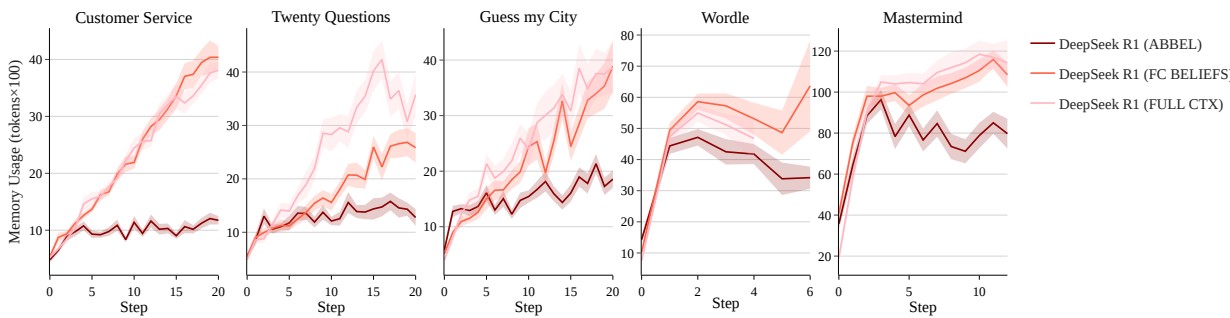

Figure 7: The memory usage at each step, defined as max(input + output tokens for belief updating, input + output tokens for action selection), representing the inference-time memory requirement. After the first few steps, ABBEL uses significantly less memory than the other frameworks.

Conditioning on belief states generated by ABBEL and FC BELIEFS rather than full histories significantly reduces action reasoning length for comparable performance in several environments. We find that this is because the reasoning models naturally integrate information from the interaction history as the first step of reasoning, and access to beliefs allows them to skip this part of the reasoning process. Concretely, controlling for performance by only averaging over environments where the success rate under ABBEL was on par with FULL CTX, DeepSeek R1 used 62% fewer reasoning tokens for action selection, and 24% fewer total reasoning tokens per step. Gemini 2.5 Pro generally used significantly fewer reasoning tokens than DeepSeek R1 across all rollout frameworks, and ABBEL further decreased its action reasoning tokens by 24%, and only increased the total reasoning tokens per step by 48% despite making twice as many calls as FULL CTX.

We also find ABBEL often uses even less reasoning than FC BELIEFS while achieving similar success rates (e.g., Deepseek R1 in *Twenty Questions*, *Guess my City* and *Mastermind*). Inspecting the reasoning traces (see Appendix C.2 for examples), we find that R1 has a strong prior to ignore the belief state and reconstruct a posterior from the interaction history when available, so using belief states as a bottleneck provides an additional benefit of preventing unnecessary extra reasoning over histories when beliefs are sufficient.

Accounting for all tokens involved at each interaction step, including input contexts, output reasoning, beliefs, and actions, we find ABBEL uses fewer tokens and requires less memory in most environments once the interaction exceeds 5 steps (see Figures 6 and 7).

## C.2 Unnecessary belief state reconstruction from the interaction history

We highlight how frontier models wastefully reconstruct previously stated information in the FULL CTX and FC BELIEFS settings.

We find that Deepseek R1 reconstructs a posterior belief from the full interaction history from scratch, as the first step of reasoning about the next action:

---

**Deepseek R1 (Full Ctx) Action Selection Reasoning**

Okay, let's see. So, the history so far has three guesses: `CLOUD`, `PULSE`, and `FAULT`. Let me recap the feedback from each.

- **From CLOUD:**
  `C`, `O`, `D` are not in the target. `L` (second letter) is present but wrong position. `U` (fourth letter) is present but wrong position.

- **From PULSE:**
  `P`, `S`, `E` are not in the target. `U` (second letter) and `L` (third letter) are present but wrong positions.

---

> - **From `FAULT`:**
>   `F, A` are not in the target. `U` (third), `L` (fourth), and `T` (fifth) are present but wrong positions.
>
> Putting this together, the letters that are definitely in the target word are `L`, `U`, and `T`. Because in `FAULT`, even though `F` and `A` are excluded, `U, L, T` are confirmed. Additionally, from `CLOUD`, `L` and `U` are confirmed but not in positions 2 and 4 respectively. From `PULSE`, `U` and `L` are again confirmed but positions 2 and 3 are wrong.
> . . .

We find that Deepseek R1 with belief prompting sometimes fails to leverage the explicit belief state in its context, and wastefully reconstructs a posterior belief from the interaction history:

> **Deepseek R1 (FC beliefs) Action Selection Reasoning**
>
> **Belief State**
> First letter is `S`. Excluded letters: `P, O, I, L, E`. Correct letters: `A` in position 3, `R` in position 4, `T` in position 5. Target word format: `S_ART`. Possible letters for position 2: `B, C, D, F, G, H, J, K, M, N, Q, U, V, W, X, Y, Z`.
>
> **Reasoning**
> Okay, so I'm trying to figure out the next guess for Wordle. Let's recap what we know so far.
> The first guess was `SPOIL`. The response told us that `S` is correct in the first position. The letters `P, O, I, L`, and `E` (from the fifth letter in `SPOIL` and the fifth in `STARE`) are excluded. Then the second guess was `STARE`, which confirmed that `S` is still correct in position 1. The letter `T` is present but not in position 2. Also, `A` is correct in position 3, `R` in position 4, and `E` is excluded. Now the target word structure is `S _ A R T`. Wait, no, wait. Wait, let me check again.
> Wait, in `STARE`, the fifth letter was `E`, which is excluded. But the response for `STARE` said that the fifth letter `E` is not in the target word. So the target word structure is `S _ A R _`, since the fifth letter isn't `E`. Wait, hold on. No, the correct letters from `STARE`: third letter `A` (position 3), fourth letter `R` (position 4). The fifth letter `E` is excluded, so the fifth position can't be `E`. But the first letter is `S`. The second letter is `T` in `STARE`, but it's in the target word in a different position. So `T` exists but isn't in position 2. So the target word has `T` somewhere else. But the structure we have is `S _ A R _`, with the fifth letter not `E`, but from the first guess `SPOIL`, the fifth letter `L` is excluded. So fifth position can't be `L` or `E`. Wait, but what about the third guess? Wait, the user only provided two guesses: `SPOIL` and `STARE`. Then the belief state says the target word is `S _ A R T`. Wait, how did `T` get into the fifth position? Because in `STARE`, the fifth letter `E` was wrong, but the correct letters in `STARE` are `S` (position 1), `A` (3), `R` (4). Then `T` in position 2 is present but in a different spot. So `T` must be in position 5? Because the only other positions are 2 (which was `T` in `STARE` but it's not there), or 5. Because positions 1, 3, 4 are already `S, A, R`. So `T` must be in position 5. That makes the target word `S _ A R T`. So fifth letter is `T`. So structure is `S _ A R T`.
> . . .

Finally, we find that Deepseek R1 under ABBEL reasons directly with the belief state:

> **Deepseek R1 (ABBEL) Action Selection Reasoning**
>
> **Belief State**
> Excluded letters: S, L, T, R, C, D, H, V
> Confirmed present letters: A (position 2), E (position 4), N (position 5)
> Possible letter positions:
> 1: [B, F, G, I, J, K, M, O, P, Q, U, W, X, Y, Z]
> 2: [A]
> 3: [B, F, G, I, J, K, M, O, P, Q, U, W, X, Y, Z]

> 4: [E]
> 5: [N]
>
> **Reasoning** Okay, let's see. I need to figure out the best word to guess for the third attempt in Wordle. The excluded letters are S, L, T, R, C, D. So I can't use any of those. The confirmed letters are A in position 2 and E in position 4. So the word must be _ A _ E _.
> My first thought is to maximize the information gain. Since this is the third attempt and there are three more left, I shouldn't be too greedy yet. I need to test possible letters in the unknown positions (1, 3, 5) while using letters that aren't excluded.
> Looking at the possible letters for positions 1, 3, and 5: they include B, F, G, H, I, J, K, M, N, O, P, Q, U, V, W, X, Y, Z. I should choose a word that uses as many of these as possible in the unknown slots to check which ones are present.
> . . .

### C.3 Hallucinating Past Observations During Belief Update Reasoning

We provide an example ABBEL belief update from Deepseek R1 in Mastermind, demonstrating how LLMs may hallucinate observations from previous steps when reasoning over a belief state in multi-step environments.

> **Game State**
>
> **Secret:** 4 5 1 8
> **Step:** 5 out of 12
> **Goal Reached:** No

> **Prior Belief**
>
> **Exclusions**: 0, 6, and 9 are not in the code.
> **Possible Codes**:
> - 4 5 [1/2/3] [7/8]
> - 4 5 [7/8] [1/2/3]

> **Action and Observation**
>
> **Action:** 4 5 1 7
> **Observation:** 3 exact matches, 0 partial matches

> **Deepseek R1 (ABBEL) Belief Update Reasoning**
>
> ...but previous feedback said that the third guess (4 5 7 8) had a partial match...

4568 was never guessed; this hallucinated previous feedback leads R1 to conclude that 7 was the partial match and must be in position 4, ruling out the true code 4518 from the posterior:

> **Updated Belief State**
>
> **Exclusions**: 0, 1, 6, 8, 9 are not in the code.
> **Possible Codes**:
> - 4 5 2 7
> - 4 5 3 7

Table 4: Training settings used across environments. The mini batch at every gradient update step was set to the number of tensors present in the step to prevent off-policy updates, which have been shown to result in training instability in Qwen models.

| Name | QA | ColBench | Combination Lock |
|---|---|---|---|
| Optimization Algorithm | GRPO | GRPO | GRPO |
| AdamW learning rate | 1e-7 | 1e-7 | 1e-7 |
| batch_size | 16 | 16 | 16 |
| GRPO n rollouts | 2 | 2 | 2 |
| mini_batch | N/A | N/A | N/A |
| training_steps | 260 | 100 | 140 |
| num_epochs (calculated equivalent) | 3.2 | 0.16 | 3.11 |
| Learning rate decay | 0.0 | 0.0 | 0.0 |
| Gradient clipping | 1.0 | 1.0 | 1.0 |

Table 5: Approximate Training Overhead (Qwen2.5-7B-Instruct).

| Model | Environment | 4-A100 PCI Hours |
|---|---|---|
| ABBEL | Combination Lock | 30 |
| ABBEL-dom-BG | Combination Lock | 28 |
| ABBEL | Multi-Objective QA | 24 |
| ABBEL-PBP | Multi-Objective QA | 20 |
| ABBEL | ColBench | 20 |
| ABBEL-rec-BG | ColBench | 12 |

## D   Reinforcement Learning Experiment Details

We train using the VeRL-agent library (Feng et al., 2026) across all our RL experiments. See Table 4 for the hyperparameters used, which were identical across environments except for the number of training steps and the resulting equivalent number of epochs.

See Table 5 for the approximate training time for each model and environment. The shorter training time for ABBEL-dom-BG compared to ABBEL in Combination Lock is a consequence of the domain-specific belief grading producing shorter belief states and fewer steps per rollout (the agent behaves more optimally, finding the secret code in fewer guesses on average). ABBEL-PBP also has a lower training overhead than ABBEL due to the belief length penalty causing the model to generate shorter belief states. Domain-general belief grading allows ABBEL-rec-BG to train in half as many steps as ABBEL in ColBench, though it results in somewhat longer belief states, resulting in the training overhead being reduced by a factor of about 5/3.

### D.1   Combination Lock

*Combination Lock* has the same feedback dynamics as *Wordle* with 3-character codes and guesses, while additionally enforcing that all three characters of the secret code and of every guess must be unique. Unique secret codes of 3 vocabulary characters were sampled, with a larger disjoint vocabulary and increased horizon at test time (see Table 6).

> **Combination Lock Game Instructions**
>
> You will determine the correct combination of characters at [Position 1, Position 2, Position 3] in a 3-character combination lock through iterative reasoning and queries.
> All 3 characters are unique.
> The set of valid characters are as follows: ['0', '1', '2', '3', '4', '5', '6', '7', '8', '9']

Table 6: Characteristics of the *Combination Lock* environments.

| Setting | Horizon (H) | Vocabulary | Answer Space Size |
|---------|-------------|------------|-------------------|
| Train | 12 | 012345689 | 720 (3 unique digits) |
| Test | 16 | qawsedrftgyhujik | 3360 (3 unique letters) |

> Each action is a query of the form ['char 1', 'char 2', 'char 3'].
> Each time you query a combination, you will get feedback from the user about each character: either not in the combination, in the combination but in a different position, or in the combination and in the right position.
> You can make up to 12 queries.
> Your goal is to find the correct combination in the least number of queries.

We prompted Qwen2.5-7B-Instruct to first think step by step between `<think>...</think>` tags, and then generate actions or beliefs between `<action></action>` or `<belief>...</belief>` tags.

> **Combination Lock Belief Prompt**
>
> Now update your belief state to include all important new information you have gathered. Do not say anything about future actions. Think step by step and then output your new belief state inside `<belief> ... </belief>`, e.g., `<think>Any thinking</think><belief>your new beliefs</belief>`.

> **Combination Lock Action Prompt**
>
> Now think step by step and then output your next action formatted as a list of 3 characters inside `<action> ... </action>`, e.g.,`<think>Any step by step, short and concise thinking to determine your next action</think><action>['char 1', 'char 2', 'char 3']</action>`.

Invalid generations did not count as an environment step, i.e. did not impact regret, but we limited the number of generation calls per game to $H$ (FULL CTX) or $2H$ (ABBEL and FC BELIEFS); see Table 7 for details. Each trajectory ends in success once the secret code is guessed, or failure if either the generation limit or environment horizon is exceeded, with reward defined as follows to encourage succeeding with as few guesses as possible:

$$\mathcal{R} = \begin{cases} (H + 1 - \text{environment steps taken})/H & \text{if trajectory successful} \\ -1 & \text{otherwise.} \end{cases} \quad (7)$$

See Table 4 for the training settings and hyper parameters used, and Algorithm 1 for the belief grading algorithm.

**Combination Lock Belief Grading Details**

> **Belief State Parsing Prompt for Combination Lock Grader**
>
> Your job is to parse belief states generated by another langauge model while its being trained. The other langauge model is playing a combination lock game. Here are the rules to that game:
> `Combination_Lock_instructions`
> The problem is I can only parse a very simple format:
> "
> possible characters in position 1: []
> possible characters in position 2: []
> possible characters in position 3: []

Table 7: Handling of invalid generations in *Combination Lock*.

| Case | Description | Outcome |
|------|-------------|---------|
| Valid action | The action generation is correctly formatted as `<action>[c1, c2, c3]</action>` with three unique characters. | Both generation and environment steps are incremented, and feedback is presented in a newline separated list. e.g,:
`8 is in Position 1!`
`6 is not in Position 2, but is in the lock`
`9 is not in the lock` |
| Invalid action | Most often errors take the form of `[action>...</action>` or repeated characters. | Generation step is incremented, and the model receives a message stating the action is invalid, reiterating the required format and prompting regeneration. |
| Invalid belief | Not using `<belief></belief>` tags. Errors tend to result from forgotten beginning/ending angle brackets or misspellings of `belief`. | Generation step is incremented, and the model receives a message stating the belief is invalid, reiterating the required format and prompting regeneration. |

---

"
Here are some examples
if you see:
"
Position 2 is '1'. Possible characters for Position 1 and Position 3 are ['3', '4', '5', '6', '7', '8', '9'].
These characters must be unique.
"
You should return:
"
possible characters in position 1: [3, 4, 5, 6, 7, 8, 9]
possible characters in position 2: [1]
possible characters in position 3: [3, 4, 5, 6, 7, 8, 9]
"
...
Please parse the following

---

**Algorithm 3** Belief Grader Function for Combination Lock

---

**Require:** belief parsing prompt $p_p$, belief parser model $\Pi$.
  **function** $f_{\text{BG}}(b_{t+1}, \text{traj}, t)$
    $b_{t+1}^* \leftarrow \text{compute\_possible\_digits}(\text{traj}[:t])$    ▷ Compute the possible digits at each position in the lock.
    $\text{parsed\_belief} \sim \Pi(\cdot | p_p, b_t)$                    ▷ Parse $b_{t+1}$ into the same format as $b_{t+1}^*$.
    **return** $\text{parsed\_belief} == b_{t+1}^*$                  ▷ Return reward of 1 if $b_{t+1}$ is correct.
  **end function**

---

**Example Belief States in Combination Lock**

Early on in training belief states were messy and difficult to parse:

---
**Belief Update Prompt**

`Combination_Lock_instructions`
Your current belief state: `<belief>`No prior belief.`</belief>`

---

Your last action:
`<action>`['0', '1', '2']`</action>`
Environment feedback:
0 is not in Position 1, but is in the lock
1 is not in the lock
2 is not in the lock
Now update your belief state to include all important new information you have gathered.
Do not say anything about future actions. Think step by step and then output your new belief state inside `<belief>` ...  `</belief>`, e.g., `<think>`Any thinking`</think>``<belief>`your new beliefs`</belief>`.

---

**Response**

`<think>`Given the feedback from the query['0', '1', '2']:
- '0' is not in Position 1 but is in the lock.
- '1' and '2' are not in the lock.
This means '0' must be in either Position 2 or Position 3, and the characters for Position 1, 2, and 3 are now limited to ['0', '1', '2'] but with '1' and '2' not in Position 1. Therefore, '0' is the only character that can be in Position 1, Position 2, or Position 3 excluding the position it is not in.
So, Position 1 cannot be '0', and it must be one of '1' or '2'. Positions 2 and 3 can be either '0', '1', or '2', but '1' and '2' cannot be in Position 1.`</think>`
`<belief>`Position 1 can be '1' or '2'.
Position 2 and Position 3 can be '0', '1', or '2', but '1' and '2' cannot simultaneously be in Position 1 and the other positions at the same time.
'0' must be in either Position 2 or Position 3.`</belief>`

---

Near the end of the belief state grading training run, we find belief states become more parse-able, and the model is more able to handle complex posterior updates.

---

**Prompt**

game_instructions
Your current belief state: `<belief>` given that '2' cannot be in position 3, it must be in position 1 or position 2. the valid characters are now ['2', '3', '4', '5', '6', '7', '8', '9']. '0' and '1' are not in the combination. the remaining characters need to be tested to determine their correct positions.`</belief>`
Your last action:
`<action>`['2', '3', '4']`</action>`
Environment feedback:
2 is not in Position 1, but is in the lock
3 is not in the lock
4 is not in the lock
Now update your belief state to include all important new information you have gathered.
Do not say anything about future actions. Think step by step and then output your new belief state inside `<belief>` ...  `</belief>`, e.g., `<think>`Any thinking`</think>``<belief>`your new beliefs`</belief>`.

---

**Response**

`<think>`From the feedback, we know:
- '2' is in the combination but not in Position 1, which means '2' must be in Position 2.
- '3' is not in the combination, so we can exclude it from the valid characters.
- '4' is not in the combination, so we can exclude it from the valid characters.

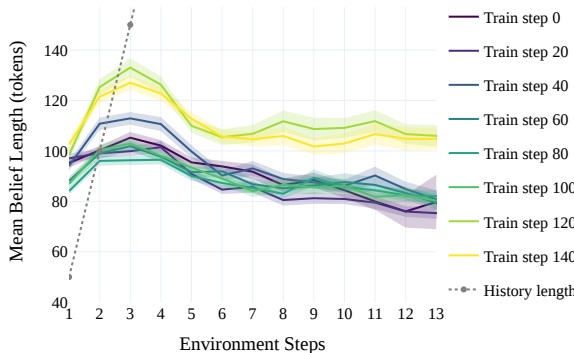

Figure 8: ABBEL belief state lengths over training in *Combination Lock* without belief grading. The model learns to generate longer beliefs throughout training, but they still remain significantly shorter than the interaction history.

| Model | Belief In | Belief Out | Action In | Action Out | Total In | Total Out |
|-------|-----------|------------|-----------|------------|----------|-----------|
| **ABBEL-Zero** | 1317.6±3.4 | 189.6±1.7 | 673.8±2.3 | 76.7±0.5 | **1991.4±4.1** | **266.3±1.8** |
| **ABBEL-PBP** | 1181.2±22.3 | 117.0±8.4 | 618.9±26.5 | 88.6±43.1 | **1800.1±34.6** | **205.6±43.9** |
| **MEM1-Instruct** | N/A | N/A | 1380.6±20.8 | 196.2±6.0 | **1380.6±20.8** | **196.2±6.0** |

Table 8: Average input and generated token counts for each step in 16-Objective QA.

---

The valid characters are now further reduced to ['2', '5', '6', '7', '8', '9']. Since '2' is fixed in Position 2, the remaining characters need to be tested in both Position 1 and Position 3 to determine their correct positions.**</think>**
**<belief>**Given the feedback, '2' is fixed in Position 2. The valid characters are now ['2', '5', '6', '7', '8', '9']. '3' and '4' are not in the combination. The remaining characters need to be tested to determine their correct positions in Position 1 and Position 3.**</belief>**

---

### D.2  Multi-Objective QA

See Table 4 for the training settings and hyper parameters used.

---

**QA Belief Prompt**

Now update your belief state to be a concise summary of all essential information you have gathered. Do not say anything about future actions. Think step by step and then output your new belief state inside `<belief> ... </belief>`, e.g., `<think>Any thinking</think><belief>your new belief</belief>`.

---

**QA Action Prompt**

Now think step by step and then output your next action formatted as `<think> ... </think><search> ... </search>` or `<think> ... </think><answer> ... </answer>`. Remember if it is your last step you must answer. You have `<H-t+1>` steps remaining.

---

**Peak Belief Penalty Details**

To calculate the penalty for each trajectory, we take the token count of longest belief state in the trajectory, subtract the mean over all trajectories in the batch, and apply a 0.01 scaling factor. We only apply peak belief penalties to trajectories which do create a valid belief state, so as not to reward generating empty

Table 9: Multi-objective QA results (arrows indicate desired directions) with memory models listed in the bottom section. Training with peak belief penalties (PBP) lets ABBEL surpass MEM1 on both performance and memory efficiency and perform close to FULL CTX for more than 2 objectives. We report the mean and SEM over 3 seeds for all trained models. To help situate ABBEL's performance, we also include metrics for a representative external memory retrieval method, A-MEM (Xu et al., 2025), applied to Qwen2.5-7B-Instruct. Numbers for MEM1 Base and A-MEM taken from Zhou et al. (2026).

| Model | 2-Objective | | 8-Objective | | 16-Objective | |
|---|---|---|---|---|---|---|
| | EM Score↑ | Tokens×$10^2$↓ | EM Score↑ | Tokens×$10^2$↓ | EM Score↑ | Tokens×$10^2$↓ |
| FC Zero | 0.30 | **11.25** | 0.37 | **16.06** | 0.40 | **15.40** |
| FULL CTX | **0.80±0.02** | 18.90±0.65 | **2.69±0.11** | 67.75±3.50 | **3.63±0.27** | 96.13±2.11 |
| A-MEM | 0.29 | 14.10±0.10 | 1.13 | 18.6±0.1 | 0.73 | 18.80±0.14 |
| MEM1 Base | 0.71 | **6.40±0.02** | 1.87 | 8.01±0.06 | 1.97 | 10.40±0.09 |
| MEM1 Instruct | **0.75±0.01** | 7.12±0.25 | 1.80±0.22 | 9.61±0.29 | 2.47±0.27 | 10.86±0.20 |
| ABBEL Zero | 0.53 | 6.85 | 1.28 | 8.67 | 1.62 | 9.46 |
| ABBEL | **0.73±0.01** | 6.94±0.09 | **2.34±0.06** | 9.15±0.18 | **3.44±0.10** | 10.39±0.28 |
| **ABBEL-PBP** | 0.67±0.03 | **6.32±0.23** | 2.13±0.07 | **7.13±0.49** | **3.45±0.05** | **7.29±0.32** |

beliefs. In addition, we set the penalty to 0 if the length does not exceed a fixed threshold of 400 tokens for either trajectory in the group, we do not normalize the lengths by the in-batch range, and we apply the penalty after advantage normalization, such that as the belief states get shorter the penalty has a smaller impact. We found this was important to avoid over-compressed beliefs which significantly harm performance.

We note that it might be more effective to account for how many steps the agent took when calculating PBP, because longer belief states may be required to describe the information collected over more steps of information gathering.

**Example Belief States in 16 Objective QA**

---

**ABBEL**

**Questions**
The principal author of the US Constitution and writer of over 1/3 of the Federalist Papers, which US President, the 4th, was CIC during the War of 1812?; Of what modern country is Sarajevo the captial of?; Pre restraining order(s), who did People magazine name as their first "Sexiest Man Alive", in 1985?;... What 70's group released a statement on Monday that read, in part "We are deeply dismayed by today's announcement from the YMCA that they feel a name change and a rebranding are in order after 166 years"?

**Belief State**
`<belief>`James Madison was the 4th US President and was CIC during the War of 1812. We need to find the "Sexiest Man Alive" in 1985, the part of the human body the carotid arteries supply, the innovation introduced by the Chillicothe Baking Company, the hand holding the torch on the Statue of Liberty, the sense referred to by the term gustatory, the canal construction started on July 4, 1817, the baseball legend who retired in 1939, the 2007 movie based on the plot summary, the Latin phrase for something designed for a specific problem or task, the planet with Phobos and Deimos as moons, the number of blackbirds in the nursery rhyme, the proverb, the two cities in Charles Dickens' novel, and the 70's group that released a statement about the YMCA.`</belief>`

---

MEM1 does not separate the belief from the reasoning, which can lead to longer internal states.

---

**MEM1-Instruct**

**Questions**
Who is James Cecil, 6Th Earl Of Salisbury's paternal grandfather?; ... Where did the director of film The Woman In The Hall study?; What is the place of birth of the composer of film Chakravalam Chuvannappol?; Where did Africanus Fabius Maximus's father die?; Which film has the director who is older than the other, Sign Of The Pagan or Spawn Of The North?

**Internal State**
`<think>`From the documents, we can infer that Jacques Becker, the director of "Edward And Caroline," was born on 15 September 1906 in France. The place of birth of the composer of "Chakravalam Chuvannappol" is not provided, so we will need to search for more information about the composer of this film. The father of Africanus Fabius Maximus is not provided, so we will need to search for more information about Africanus Fabius Maximus. The director of "The Woman In The Hall" is not provided, so we will need to search for more information about the director of this film. The place of death of Anne Devereux's husband is not provided, so we will need to search for more information about Anne Devereux. The directors of "Friday The 13Th (1916 Film)" and "Beaumarchais (Film)" are not provided, so we will need to search for more information about these directors. The film "Sign Of The Pagan" was released in 1935, and "Spawn Of The North" was released in 1934. Therefore, the director of "Sign Of The Pagan" is older than the other.`</think>`

---

Training ABBEL with peak belief penalties leads to even more concise beliefs that do not enumerate all the unknown information:

---

**ABBEL-PBP**

**Questions**
What literary style describes the 1968 book that features a 1939 international Harvester school bus?; Is the building located at 200 West Street taller than the one at 888 7th Avenue?; ... Are Waste Land and Shikashika both types of drinks?

**Belief State**
`<belief>`The 1968 book "The Electric Kool-Aid Acid Test" by Tom Wolfe is written in the New Journalism style. The building at 110 West 7th Street is taller than the one at 888 7th Avenue.`</belief>`

---

### D.3 ColBench

See Table 4 for the training settings and hyper parameters used.

---

**ColBench Belief Prompt**

Now update your belief state to include all important new information you have gathered. Do not say anything about future actions. Think step by step and then output your new belief state inside `<belief>` ... `</belief>`, e.g., `<think>Any thinking</think><belief>your new beliefs</belief>`.

---

**ColBench Action Prompt**

Now think step by step and then output your next action formatted as `<think>` ... `</think><ask>` ... `</ask>` or `<think>` ... `</think>` ... ``. Remember if it is your last step you must code. You have `<H-t+1>` steps remaining.

---

Below we show how we construct the context provided to the agent model, to calculate its predicted probability of the tokens in the last observation $o_t$ (i.e., environment feedback) conditioned on the task instructions and $b_t, a_t, b_{t+1}$.

---

**ColBench Domain-General Belief Grading Context Template**

Global Instruction: `ColBench_instructions`
Your new belief state is: `<belief>new_belief</belief>`
Your past belief state was: `<belief>prior_belief</belief>`
Your past action: `<action>last_action</action>`
Your past environment feedback: `<environment>`

---

**Algorithm 4** Information Reconstruction Belief Grader Function used for ColBench

---

**Require:** Current language model $\pi_\theta$
   **function** $f_{\text{BG}}(b_{t+1}, \text{traj}, t)$
      context $\leftarrow$ apply_template(traj$[b_t, a_t], b_{t+1}$)      ▷ Condition on $b_{t+1}$ and the prior belief and action.
      obs $\leftarrow$ traj$[o_t]$      ▷ Get the observation $o_t$ that $\pi_\theta$ updated $b_t$ on to generate $b_{t+1}$.
      score $\leftarrow \sum_{i=1}^{|\text{obs}|} \log \pi_\theta(\text{obs}_i \mid \text{obs}_{<i}, \text{context})$      ▷ Sum logits to get ability of $\pi_\theta$ to reconstruct $o_t$.
      **return** max(score, $-0.9$)      ▷ Cap the score to prevent over-optimization.
   **end function**

---

We provide examples of belief states generated after training with and without belief grading, showing that grading based on information reconstruction resulted in more information being retained in the belief state.

---

**Example ABBEL-rec-BG Belief State**

The function will assign a score of 5 for the phrases 'MUST BUY!', 'Brilliant!', and a score of 2 for each occurrence of 'BUY' in a review. It will assign a score of 4 for the phrase 'Almost!', and 1 for '1 (100% helpful)'. Reviews containing positive indicators like 'MUST BUY!', 'Brilliant!', or 'BUY' will contribute to the overall score. The scoring system will sum these scores, and any review that does not contain any of these indicators will have a score of 0.

---

**Example ABBEL (No Belief Grading) Belief State**

Target year: 2050, Reduction percentage: 50%, Current emissions data: symbolic variables (e.g., *current_emissions*), Clarification needed: total emissions cut by 2050 or annual reduction rate.

---

