# OpenReview forum: "ABBEL: Learning Natural-Language Belief States for Memory-Efficient Interaction"
_TMLR — Under review for TMLR_

### Review · Reviewer_wDii · 2026-07-07

**Summary Of Contributions:**

The paper proposed a framework how to use the interaction history (a long trace) by converting them into a belief rather than verbatim context, to achieve effective reasoning with shorter but informative context.

Strengthes:
1. The paper tackles an important problem of multi-step planning, when previous planning history is existed, how to present the data to model to assure the data is used more effectively.
2. The paper presented an approach to curate the data for further reinforcement learning to boost the performance instead of relying on the human-collected preference data.

Weaknesses:
1. Lacks of comparison on simple baselines and related works on how the previous work handles the long context (multi-step scenarios)
2. The writing is confusing, as of now, the authors split the proposed sections into two parts (3 and 4) and evaluation of each part (5 and 6), but I am confusing that why the method in sec 3 can not be applied to the open-weight model used in RL? It will be better to get the holistic view the improvement with and without finetuning.

**Additional Comments:**

Does figure 2 contain the token usage during belief generation?

**Audience:**

Yes

**Audience Explanation:**

How to use context efficient and effectively is one of the key research topics.

**Claims And Evidence:**

Yes

**Claims Explanation:**

But when using along with the model as (sec 4) is, the improvement seems to not clear and no comparison with other methods.

**Requested Changes:**

1. As described in the related work section, there are many works are handling long context; however, none of them are compared, please add the comparison.

2. Even though the belief is compacted but it does not always show the improvement over full context; moreover, what is the total token usages for the proposed method to the full context method? E.g., the proposed method need to generate belief by feeding the previous belief, action and observation?

3. How should the users to involve the techniques in practice? In section 5, two methods for RL are proposed but they are evaluated separately, can't they be used simultaneously? Or the users need to have strong domain knowledge to configure the algorithm properly?

4. I might be confused the proposed methods to a naive baseline, in practice, when context becomes longer and longer, we will ask LLM to summarize it before feeding as a full context, is it can be a strong baseline to compare? Is the uniqueness of the proposed method is how the prompt is crafted to generate "belief"?

---

> ### Author Response · Authors · 2026-07-12
>
> We thank reviewer wDii for taking the time to review our paper and for their generally positive review. We are glad that they agree we tackle an important problem that is of key interest to TMLR’s audience, and that our claims are supported by strong evidence. We address their questions and concerns below.
>
> >lack of comparison with prior work
>
> We compare with the prior multi-step long context work MEM1[1], noting that most other published work for condensing long contexts only apply to the single step setting, where all information is presented up front, and cannot handle the multi-step setting with dynamically growing contexts and active information gathering.
> If there is a particular multi-step long context work the reviewer has in mind, we would appreciate if they could point us to it!
>
>
> > lack of comparison on simple baselines: when context becomes long, we will ask LLM to summarize it. can this be a strong baseline to compare? Is the uniqueness of the proposed method how the prompt is crafted to generate "belief"?
>
> We clarify that this corresponds to the ABBEL-Zero baseline, which prompts the LLM to summarize the context (by generating a belief state) but does not do any RL fine-tuning.  ABBEL’s uniqueness and improved memory efficiency mainly come from the belief state supervision (Belief Grading and Peak Belief Penalties), not the belief prompting by itself.
>
> >why the method in sec 3 can not be applied to the open-weight model used in RL?  It will be better to get the holistic view the improvement with and without finetuning.
>
> By " the method in sec 3" do you mean the ABBEL rollout framework described in section 2 that we evaluate with frontier models in section 4? We do use that same rollout framework when training the open-weight models: throughout the paper, all models with ABBEL in the name are using the ABBEL rollout framework. The improvement with and without finetuning can be seen by comparing the ABBEL models with ABBEL-Zero (i.e., zero-shot performance, no finetuning) throughout the experiments in section 6. We will edit the manuscript to make this clearer.
>
> >But when using along with the model as (sec 4) is, the improvement seems unclear
>
> We would like to clarify that, as stated at the top of section 4, this section identifies the failure modes of prompting LLMs to perform recursive self-summarization. Thus, section 4 should not be taken as a demonstration of the efficacy of our methods- instead it *motivates* the supervision methods we introduce in section 5.
>
> >Even though the belief is compacted it does not always show the improvement over full context
>
> It is expected that summary models will not outperform full context models in our experiments, because summary models have access to less information. The goal is memory efficiency, not higher absolute performance than the full context setting: our methods use an order of magnitude less memory for a relatively small drop in performance. And our methods surpass prior memory agent work (MEM1) on both performance and memory usage.
>
> >Does figure 2 contain the token usage during belief generation? what is the total token usages for the proposed method to the full context method? E.g., the proposed method need to generate belief by feeding the previous belief, action and observation?
>
> No, Figure 2 just shows the length of the full context vs the length of the belief states and does not include reasoning tokens. This is because only the belief state is used for the context at the next step, and the reasoning tokens are discarded, so the figure shows the context reduction.
>
> Figures 5 and 6 in the appendix provide more information about token usage including reasoning tokens. In particular, we plot a comparison of total token usage in Figure 6 (bottom row), showing that ABBEL has comparable usage and can even reduce the number of tokens processed at each step compared to the full context setting.
>
>
> >How should the users to involve the techniques in practice? In section 5, two methods for RL are proposed but they are evaluated separately, can't they be used simultaneously? Or the users need to have strong domain knowledge to configure the algorithm properly?
>
> The methods in section 5 address different problems. If the full context model outperforms the naive summary model (e.g., in ComboLock and Colbench), it is likely that the belief states contain errors or omissions, and Belief Grading can help reduce them. Peak Belief Penalties are useful for settings where memory is especially expensive, reducing memory use by improving the conciseness of the belief states. Thus, only basic understanding is needed to see where either will be applicable. They can be used simultaneously if both problems are present, but we evaluated them separately so it would be easier to see their effects.
>
> Many thanks again for your consideration!
>
> [1] Zhou, Zijian, et al. "MEM1: Learning to synergize memory and reasoning for efficient long-horizon agents." (2026)

---

### Review · Reviewer_pA8W · 2026-07-18

**Summary Of Contributions:**

This paper proposes ABBEL, a framework for long-horizon LLM agents that maintains recursively updated natural-language belief states instead of full interaction histories. The key idea is to separate memory (belief states) from reasoning and action generation, enabling more interpretable and memory-efficient context management. The authors further introduce two reinforcement learning techniques, belief grading and peak belief penalties, to improve belief quality and reduce memory usage.

The paper provides extensive experiments across several sequential decision-making environments, showing that ABBEL can reduce context length while maintaining or improving performance compared to prior memory-based approaches.

The main strength of the paper is the clear formulation of belief states as an explicit and trainable memory representation for LLM agents. However, the empirical validation is mostly conducted in controlled environments, and the extent to which ABBEL generalizes to more complex real-world agent tasks remains unclear.

**Audience:**

Yes

**Audience Explanation:**

This paper studies an important and timely problem in LLM agents: how to maintain efficient long-term memory. The proposed explicit belief state representation provides an interesting perspective for memory-augmented LLM systems and may benefit researchers working on long-horizon agents and context management.

**Broader Impact Concerns:**

No major broader impact concerns are identified. The authors may briefly discuss potential risks of persistent memory errors in long-running LLM agents.

**Claims And Evidence:**

No

**Claims Explanation:**

No. The experiments provide evidence that ABBEL can improve memory efficiency and maintain performance in several controlled sequential decision-making environments. However, the evidence is not sufficient to support broader claims about its effectiveness for general long-horizon LLM agents. In particular, the evaluated environments have relatively compact and well-defined latent states, which are favorable for belief-state representations. More complex settings such as repository-level software engineering, open-ended tool-use agents, or scientific workflows are not evaluated, leaving the general applicability of the proposed memory mechanism unclear.

**Requested Changes:**

The authors should evaluate ABBEL on more realistic long-horizon agent tasks beyond the current controlled environments. In particular, experiments on repository-level software engineering tasks (e.g., SWE-bench or similar coding agents) would provide stronger evidence that belief-state memory can benefit complex tasks requiring long-term context management. Additional evaluations on open-ended planning tasks, such as travel planning or tool-use agents, would also help assess the generality of the proposed approach.

---

### Review · Reviewer_uLgU · 2026-07-21

**Summary Of Contributions:**

**SUMMARY**

This paper describes ABBEL, a framework for memory-efficient multi-step LLM agents based on maintaining a recursively-updated natural-language belief state, instead of the full interaction history. The framework separates belief-updating from action-selection into two distinct LLM calls per step, allowing ABBEL to grade the belief, penalize its length, and force it to be the real bottleneck.

The paper first analyzes the gap in performance of ABBEL summarization versus full-context agents on zero-shot frontier models (Gemini/Deepseek), and attributes failure modes to 1) belief omissions and update errors that propagate forward, and/or 2) inefficient verbosity.
Based on these findings, it applies two GRPO-trained RL fixes: 1) Belief Grading (BG), an additional GRPO task that re-samples beliefs from the same context and grades them (with two options: domain-specific or domain-general); 2) Peak Belief Penalties (PBP), which penalizes the length of the peak (longest) belief state in a trajectory.

The framework claims 40% higher performance while using 67% of memory (= 33% reduction) compared to the competing MEM1 framework which, unlike ABBEL, entangles memory and reasoning in one call.

**STRENGTHS**

- paper is well structured, with an initial analysis that identifies failure modes, followed by a targeted fix
- disentangling belief from reasoning in the proposed way is a novel, meaningful contribution that makes the memory object interpretable
- reported experimental results support applicability of proposed fixes: 1) domain-specific Belief Grading (dom-BG) beats full context while using less memory; 2) reconstruction-based Belief Grading (rec-BG), the more general method, narrows the gap vs full context while using less memory; 3) ABBEL-PBP surpasses MEM1 performance (the claim in the abstract) where PBP cuts memory with minimal performance loss

**WEAKNESSES**

- there is a substantial gap between motivations and tested regime: introduction motivates the work mentioning "decision-making tasks (...) hundreds or thousands of steps of environment interaction", but no ABBEL experiment exceeds 20 steps. What are the authors expectations for longer horizons? Are there data that explore such regimes?
- lack of generalization across architectures, all fine-tuning is Qwen2.5 (7B/14B). Have the authors explored other architectures?
- the performance and memory gains claimed in the abstract are obtained for a specific set of conditions (single environment: Multi-objective QA) and the comparison mixes a re-implemented baseline (MEM1-Instruct) with numbers taken from the original MEM1/A-MEM papers. The claim should be narrowed or better contextualized
- is the penalty of PBP applied uniformly in all scenarios or only when verbosity is identified as problematic? What is the impact of applying PBP when not required?
- dom-BG closes the gap with full context agents but requires a computable ground-truth statistic, so applicability is severely constrained; the general method, rec-BG, gives weaker results
- the two graders (dom-BG/rec-BG) and PBP are each tested in isolation in separate environments. No experiment explores their combination.
- by construction, ABBEL relies 2 LLM calls per step versus MEM1's single call. It is argued that total token counts remain comparable (because the extra belief call reduces action-selection reasoning, and input tokens are cheaper/parallelizable). However, two sequential generations per step would still introduce a latency and compute-cost discrepancy that the "memory efficiency" (peak-token) metric does not capture. Can the authors quantify the wall-clock/throughput impact of the doubled call count (particularly at longer horizons, if possible)?

**Additional Comments:**

n/a

**Audience:**

Yes

**Audience Explanation:**

The topic of agentic decision-making is highly relevant and timely. With the proposed framework, the paper provides novel contributions to the field.

**Broader Impact Concerns:**

not addressed in the manuscript. No concerns on my end.

**Claims And Evidence:**

Yes

**Claims Explanation:**

The paper is generally honest with claims and limitations, but there are some claims that should be narrowed (especially the performance claim in the abstract) and some experiments generalized.

**Requested Changes:**

see WEAKNESSES above